# Wang Bi's "Confucian" *Laozi*: Commensurable Ethical Understandings in "Daoist" and "Confucian" Thinking

Paul Joseph D'Ambrosio

Institute of Modern Chinese Thought and Culture, Philosophy Department, East China Normal University, Shanghai 200241, China; pauljdambrosio@hotmail.com

**Abstract:** Wang Bi's work is often used as evidence for "Confucian" interpretations and translations of the *Laozi*. Those who argue that the explicit rejections of Confucian values in chapters 5, 18, 19, and 38 should actually be read as admonishing hollow imitation and the mere appearance of Confucian morality often cite Wang Bi. Additionally, this great philosopher is normally taken as a mere commentator who simply sought to synthesize Confucian and Daoist ideas. In this paper, I will argue that Wang's project is, in fact, far more complex and nuanced. He develops his own philosophical system, which appreciates some underlying commensurability between the *Laozi* and *Analects*. Describing him as promoting a "Confucian" *Laozi* is inaccurate as he ultimately leans more heavily on "Daoist" concepts, such as "self-so" and "non-action." In short, Wang Bi develops a unique philosophical system grounded heavily in various classics, and while his commentary on the *Laozi* is taken as "Confucian," it is, in fact, far more complex.

**Keywords:** *Laozi*; Daodejing; Daoism; Wang Bi; Confucianism; Xuanxue

## 1. Introduction

The classification of pre-Qin philosophical texts into schools has long been useful in both academic and more popular settings. However, failing to appreciate nuance and/or relying too heavily on supposed opposition between these schools can easily lead to misunderstandings in interpretation and translation. Texts may be read in a more superficial manner if alignment with a particular school is overemphasized, or when the opposition between schools is overbearing, the more nuanced relationship between texts dissolves. This is especially true of the texts associated with the two most prominent schools, namely Daoism and Confucianism. Moreover, while Chinese scholarship has often entertained the idea that they can be integrated or synthesized, the overwhelming focus of contemporary discourse emphasizes difference. For example, the *Zhuangzi* 莊子 (*Book of Master Zhuang*)[1] is often taken as being opposed to the *Lunyu* 論語 (*Analects of Confucius*). There is great evidence to support this—Confucius is often mocked, harshly criticized, and his life is even threatened in this "Daoist" classic. However, there is also excellent evidence to suggest that the *Zhuangzi* and the *Analects* share many of the same concerns and even provide similar philosophical reflections (cf. Wang 2004; Wang 2010; Nylan 2017). Strict allegiance to particular "schools" has made parsing out the complex relationship between these texts a difficult task.

The *Laozi* 老子 (*Book of Master Lao*) or *Daodejing* 道經 (*Classic of the Way and Virtuosity*) presents us with more difficulties when attempting to interpret it as promoting ideas essentially distinct from those found in the *Lunyu*. Like the *Zhuangzi*, there are passages that seemingly contain explicit and absolute rejections of Confucian values. However, given the various theories about the relationship between the persons Laozi and Confucius, as well as potential overlaps between general philosophical attitudes in the *Laozi* and *Analects*, many scholars have found avenues for dismissing the potential censure of Confucianism in the *Laozi* as actually an appeal to double-down on true Confucianism as opposed to

false appearances. There are also scholars who argue against these readings (cf. Lin 1948; Moeller 2007). Support for the Confucian interpretations and translations of the *Laozi* are often traced back, in one way or another, to the version passed down from Wang Bi 王弼 (d. 249), as well as his accompanying commentary.

In terms of ethics, virtues, and values, Wang Bi's 王弼 (d. 249) work on the *Laozi* is normally read as mainly "Confucian."[2] Wang himself invited this understanding when he revered Confucius as a higher sage than Laozi:

> "The sage (Confucius) embodies non-being, but non-being cannot be explicated, and therefore he said nothing (about it). Laozi was one (who was fettered in) this realm of being, and thus always talked about that in which he was insufficient". (Ziporyn 2003, p. 23)

Accordingly, the Neo-Daoist or Xuanxue (*xuanxue* 玄學) prodigy is taken as, at worst, merely reinterpreting the *Laozi* through a Confucian lens—or at best, synthesizing the Daoist classic with Confucius. As Brook Ziporyn summarizes, "one of his central philosophical tasks: to harmonize 'Ruism' and 'Taoism,' social norms and spontaneity, indeed to unify them."[3] (Ziporyn 2003, p. 18). Ronnie Littlejohn expresses a similar idea, saying Wang was "a self-identified Confucian" who "wanted to create an understanding of Daoism that was consistent with Confucianism." (Littlejohn n.d.). Eric Nelson similarly writes: 'Wang Bi should be considered a Confucian who "fashionably" incorporated Daoist elements because of their historical importance after the crisis of Confucian orthodoxy during the post-Han dynasty period." (Nelson 2020, p. 288)[4]. A host of Chinese thinkers, who will be mentioned below, agree with these general points.

Tang Yongtong 湯用彤 (d. 1964) explores the terrain through referencing "self-so" (*ziran* 自然), which is a major concept in the *Laozi*, as coming to represent an ethical orientation that resists the codifying potentiality of the "teaching of names" (*ming jiao* 名教). During the Han period, values and virtues related to Confucianism were institutionalized (thus referred to as a "school of names") (cf. Tang 1957). Gradually, they became largely hollow. "Village worthies" ran rampant. Several scholars, including Wang Fu 王符 (d. 163 CE) and Xu Gan 徐干 (d. 217 CE), warned that with the hypocrisy associated with Confucian virtues and "moral character" (*de* ) would come social and political confusion (cf. Makeham 1994), and they were right. Many see Wang Bi's use of "self-so" as a way to preserve Confucian values without relying on names or doctrines, which could always be subject to potentially problematic standardizations. The repercussions of these evaluations are far-reaching. They give shape to an analysis of the long-standing tradition of *Laozi* scholarship in China, including contemporary works.

For most of its history, the *Laozi* was largely read with one of two major commentaries. Heshang Gong's 河上公 (1st century BCE) work was popular among laypeople,[5] and Wang Bi's was hugely influential for scholars. Taking the latter as a representation of a, to whatever degree, "Confucian" reading of the *Laozi* tells us that major scholarly interpretations for thousands of years have been "Confucian" as well. This also colors our perspectives on translations into other languages as well as their accompanying philosophical and religious explanations. Methods for interpreting and translating the *Laozi* itself as well as its commentaries must deal, in some way or another, with the distinction between Confucian and Daoist schools, as well as a hugely influential "Confucian" reading.

In this paper, we will present Wang Bi as a philosopher in his own right. His thought should neither be reduced to being "Confucian" any more than "Daoist." Even the label "*Xuanxue*" or "Neo-Daoist" should be approached with caution—understanding what it means and the work of other *Xuanxue* thinkers is shaped by Wang Bi's philosophy, not the other way around. His interpretation and commentary on the *Laozi,* is, in fact, an attempt to demonstrate some underlying similarities between the major concerns, theories, and concepts of Confucian and Daoist thought. The project focuses mainly on Wang Bi's supposed reinterpretation of "overtly anti-Confucian" passages in ways that make them align with Confucius, while simultaneously unequivocally upholding notions from the

*Laozi* as categorically superior to Confucian virtues and value-orientations. We will look specifically at chapters 5, 18, 19, and 38 of the *Laozi*. In doing so, we will discuss how Wang's introduction and heavy reliance on "nature" or "dispositions" (*xing* 性) and "authenticity" or "genuineness" (*zhen* 真) throughout his commentary—and especially in their respective connections to "self-so" (*ziran* 自然), "non-action" (*wuwei* 無為), and thoroughgoing rejections of selfishness (*ji* 己) and desires (*yu* 慾)—makes the *Laozi* commensurable with Confucius (and some understandings of Confucianism). In other words, Wang Bi takes the promotion of "self-so" and "non-action"—which are understood largely in terms of returning to one's own "dispositions" and "genuineness," or else operating in accordance with the "dispositions" and "genuineness" of others, things, or the situation, and thereby rejecting selfishness and desires—as the shared (moral) goal of both Laozi and Confucius.[6] The discussion of these concepts frames this paper; Wang's unique understanding of them demonstrates what he sees as the underlying commensurability between the *Laozi* and Confucius.

The overall argument is that while Wang's work has been evaluated as a "Confucianization" or "Confucian-based synthesis," there is actually no clear evidence that this is a conscious project pursued by Wang Bi himself. It is just as likely, and perhaps even better evidenced, that Wang's reading represents an accurate account of what he really thought the *Laozi* itself was saying. Below we will explore how Wang Bi saw the *Laozi* and *Analects* as similar in rejecting selfishness and desires while at the same time advancing self-so and non-action. Drawing on this, Wang Bi's unique philosophy is not concerned with harmonizing, synthesizing, or unifying Confucianism and Daoism. As a philosopher in his own right, he elucidates what he saw as commensurable in the underlying projects of the *Laozi* and *Analects*. If we take this to be plausible, then our thoughts about the "Confucian" (and "Daoist") *Laozi* and the entire tradition of *Laozi* scholarship in China, including contemporary interpretations and translations, need to be adjusted accordingly.

## 2. Straw and Dogs

By all accounts, Wang Bi was nothing short of a genius. He dominated "pure conversation" (*qing tan* 清談) gatherings[7] and earned a reputation that allowed him an audience with the prominent scholar-official He Yan 何晏 (d. 249). Upon reading Wang's work on the *Laozi,* He Yan, some thirty-one years Wang's senior, supposedly burned his own commentary. In addition to writing and commenting on the *Laozi*—in a work that became standardized in China for nearly two thousand years—Wang wrote essays and extensive commentaries on both the *Analects* and *Book of Change* (*Yijing* 易經). These, too, are classics in themselves and have been hugely influential.[8] Impressive for someone who died before his twenty-fourth birthday.

Despite his illustrious résumé, Wang is regularly criticized for making rather dumbfounded mistakes. His commentary on chapter 5 is almost universally agreed to as harboring a ridiculous misreading, namely taking *chugou* 芻狗 or "straw dogs" as "straw and dogs." The *Laozi* chapter reads:

> Heaven and earth are not humane (*ren* 仁), they take the ten thousand things as straw dogs (*chugou* 芻狗); the sage is not humane (*ren* 仁) they take the hundred clans as straw dogs (*chugou* 芻狗). Between heaven and earth—does it not resemble a bellows! Empty but not exhausted. The more it moves the more comes out. Much speech means numerous failures—this is not as good as holding on to center. (Chen 2020, p. 72; translation modified)

The rejection of humaneness is generally taken to be a dig against Confucianism,[9] but the issue is complex and can be returned to after analyzing Wang's commentary in depth. The first part of his comments is generally accepted as perfectly hitting the mark. He writes:

> Heaven and earth allow things to follow their natural bent and neither engage in conscious effort nor production, leaving the myriad things to manage them-

selves. Thus they "are not humane." The humane have to produce, establish, employ (rules, laws, policies, institutions, etc.) and transform (the beneficiaries), exemplifying kindness and achievement. But with (these activities of) producing, establishing, employing, and transforming, people lose their genuineness (*zhen* 真). If people do not preserve their genuineness (*zhen* 真), they no longer have the capacity to uphold the full weight of their existence. (天地任自然, 無為無造, 萬物自相治理, 故不仁也. 仁者必造立施化, 有恩有為; 造立施 化, 則物失其真有恩有為, 列物不具存, 物不具存, 則不足以備載矣.). ([Lynn 1999](), p. 60; translation modified)

The next lines are comically off:

Heaven and Earth do not make the grass grow for the sake of beasts, yet beasts eat grass. They do not produce dogs for the sake of people, yet people eat dogs. Heaven and Earth take no conscious effort with respect to the myriad things, yet because each of the myriad things has what is appropriate for its use, not one thing is denied support. As long as you use kindness derived from a personal perspective, it indicates a lack of capacity to leave things to themselves. (地不為獸生芻, 而獸食芻; 不為人生狗, 而人食狗. 無為於萬物而萬物各適其所用, 則莫不贍矣. 若慧由己樹, 未足任也.). ([Lynn 1999](), p. 60; translation modified)

Strangely enough, a thinker of Wang's caliber was somehow unable to recognize that "straw dogs (*chugou* 芻狗)" is one word and not two. As the *Zhuangzi* 14.4 clearly demonstrates, "straw dogs" are the objects revered during a ritual and then cast aside afterwards. This fits perfectly with the idea of "Heaven and earth are not humane." Sometimes people are revered, and other times they are cast aside, and the sage should learn to act in a similar manner. It is almost unthinkable that Wang, whose entire approach to language in the *Laozi* rests on his reading of the *Zhuangzi,* could make such a simple error (cf. [Cai 2013]()).

If we first reserve judgment on what seems to be an obvious misreading of "straw dogs," we find out what is being said is actually quite plausible. In his essay on the *Laozi*, which some take as an introduction, Wang places primacy on "actualities (*shi* 實)" over "names (*ming* 名). "All names," he writes, "arise from forms" ([Lynn 1999](), p. 39)—or "*xing* 形" which is functionally equivalent here to "actualities." This is an obvious attack on a promotion of names that over-emphasizes their importance and ultimately leads to their being empty shells. It is the problem of hypocrisy and falsity—the village worthies Confucius warned of and the confusion identified by Wang Fu and Xu Gan. Wang Bi himself explains:

If the virtues of honesty and the uncarved block are not given prominence but the splendors of reputation and conduct are instead publicized and exalted, one will cultivate that which can exalt him in hope of the praise involved and cultivate that which can lead to it in the expectation of the material advantage involved. Because of hope for praise and expectation of material advantage, he will conduct himself with diligence, but the more splendid the praise, the more he will thrust sincerity away, and the greater his material advantage, the more contentious he will be included to be. ([Lynn 1999](), p. 39)

The "uncarved block" refers to one of the most famous images used by the *Laozi* to discuss Dao. In this essay, and throughout his works, Wang goes further than the famous first lines of the text, which speak of its ineffability. He either employs the term "*ci* 此," meaning "this," or simply refers to Dao by hints and without any signifier. Once a name is used, the "splendors" of a reputation can follow, and this, as he explains, can easily lead people further away from what is actually being spoken about. Moreover, it is the reason Wang reveres Confucius over Laozi. While Laozi wrote over five thousand characters trying to explain what Dao is, though he admits it is ineffable, Confucius never speaks of Dao and only hints at it. Unfortunately, Confucius's way is plagued by the very problem he set out to avoid. Reputation, praise, and material advantage are heaped on those who fulfill set expectations for what normative behavior looks like. Concentration thus befalls the

name, and the actuality is lost. True virtuosity is not thereby simply ignored but actually eschewed. Wang Bi makes his point clear:

> The heartfelt feelings that fathers, sons, older brothers, and younger brothers should have for one another will lose their genuineness (*zhen*). Obedience (*xiao*) will not be grounded sincerity, and kindness (*ci*) will no longer be grounded in actuality. All this is brought about by the publicizing of reputation and conduct. (Lynn 1999, p. 39; translation modified)

Looking back to his reading of "straw dogs" as "straw" and "dogs" now begins to make more sense. Read as "straw dogs," the passage would argue, according to Wang's line of thinking, that sometimes heaven and earth do "produce, establish, employ (rules, laws, policies, institutions, etc.)" and that sages should emulate this behavior. Similarly, "with (these activities of) producing, establishing, employing, and transforming, people lose their genuineness." Since heaven and earth do that (i.e., establish institutions of virtue) sometimes and other times do not, then it is okay for the sage to act the same way. Sages should also sometimes establish institutions of virtue and sometimes not. Remember, straw dogs are revered for a while, then used for kindling. Therefore, if "straw dogs" refers to these ceremonial pieces, then chapter 5 makes an extremely paradoxical and potentially problematic philosophical point. As separate "straw" and "dogs," we can understand how things are produced without being intended for certain utilitarian or instrumentalizing usages. Straw is not made *for* animals to eat any more than heartfelt feelings between family members are done *for* reputation, praise, or material gain. That these things may follow is fine, a merely accidental benefit, but to reverse our thinking and assume that this is something we can mechanically dissect and instrumentally exploit is detrimental to individuals and society.

We thereby see how this seemingly overt rejection of the Confucian value of "humaneness" becomes a much more complex issue in the hands of Wang Bi. It is taken as a critique of establishing names and doctrines to codify otherwise genuine heartfelt behaviors. The use of these institutions is seen as marking the degradation of the values they seek to promote. The alternative is to return to or simply preserve the genuineness already within people. This is the source of the heartfelt interactions which are themselves moral in a plain "uncarved" manner. They are the actualities that names distinguish. Concentration should, however, be on the actualities and not the names. Separating "straw" and "dogs" is about promoting this type of emphasis and moving away from intentionality and towards self-so. As Wang Bi comments on the last lines of chapter 5, "The more you apply conscious effort to something, the more you will fail." Revering straw dogs is no exception.

## 3. Consciousness and Fish

In his commentary on chapter 18, Wang Bi repeats his repudiation of applying conscious effort. Here, it is in the midst of a chapter that seems to reject every level of Confucian moral thinking. Alan Chan, for instance, finds substantial evidence here to suggest engagement "in a critique of some of the key ideas central to the 'Ru' or Confucian tradition." (Chan 2018). Indeed, it is difficult to suggest anything but; the chapter reads:

> When the great Dao is abandoned, there is humaneness and righteousness. When wisdom appears, there is great hypocrisy. When three family relations and six roles, lack harmony, there is filial piety and parental care. When the state and families are thrown into confusion, then loyal servants arise.[10] (Chen 2020, p. 139; translation modified)

As we will see with chapter 38 below, this section seems to suggest that Confucian values only represent a decaying Dao. In this context, "the great Dao" stands for social harmony and well-being. The passage itself can then be taken as expressly "anti-Confucian." Hans-Georg Moeller explains the problem from the perspective of the *Laozi*: "With the establishment of such 'positive' values as humanity (*ren*), righteousness (*yi*), knowledge (*zhi*), and filial piety (*xiao*), the Confucians also implicitly create their opposites."[11] (Moeller

2006, p. 69). The *Laozi* then thinks that Confucianism proposes virtues which "are nothing but ineffective remedies in a degenerated society." (Moeller 2004, p. 117). The sentiment is sometimes guarded against in interpretations and translations which are strongly influenced by Wang Bi's "Confucian" reading.

For example, Chen Guying 陳鼓應 (b. 1935) warns that the addition of the phrase "when wisdom appears, there is great hypocrisy (*zhi hui chu, you da wei* 智慧出, 有大)" should be omitted on the basis of the Guodian version. He states that "the superfluous addition of these phrases is probably the result of the influence exerted by the theories of extremist followers of Zhuangzi in the late Warring States period, who preposterously added them to the text." (Chen 2020, p. 139). As a result of this addition, Confucian values can be associated with hypocrisy, and we read the *Laozi* as harboring a "downright negation" of them. In other words, readings like Moeller's and other anti-Confucian interpretations of the *Laozi* can be substantiated only with this "superfluous" and "extremist" addition.

Wang Bi's version retains these lines, but his commentary is exactly opposed to the rejection of these values. His writing on the first two sentences focuses on establishing goodness (*shan* 善) through lacking conscious effort (*wu wei* 無為) and dispelling falsehood by not letting methods (for detecting falsity) be known. Like many of his contemporaries, Wang's thought utilizes thinking that is now referred to as "Goodhart's Law" and the "Cobra Effect" to turn even the phrase "when wisdom appears, there is great hypocrisy" into something related to Confucius. Wang writes: "When one employs methods and uses intelligence to uncover treachery and falsehood, his intentions become obvious, and the form they take visible, so the people will know how to evade them?" (Lynn 1999, p. 80). This is how Wang squares the anti-Confucian lines in the *Laozi* with pro-Confucian sentiments. Chen, and many others who argue that ideas in the *Laozi* and *Analects* mesh well together, rely heavily on Wang's "Confucian" reading.

More evidence for the "Confucian" reading of this chapter is found in Wang's reference to the *Zhuangzi* in his long explanation of the last sentences. Here this second Daoist classic is employed to demonstrate shared concerns between the *Analects* and *Laozi*, which also includes the participation of the *Zhuangzi*. Referring to the basic paradoxical logic of the *Laozi*, found most evidently in chapter 2, Wang Bi explains:

> The most praiseworthy of names are generated by the greatest censure, for what we know as praise (*mei*) and censure (*e*) come from the same gate. "The six relations" are father and son, older and younger brother, and husband and wife. When the six relations exist in harmony and the state maintains good order all by itself, no one knows where the obedient (*xiao*) (child, younger brother, wife) and the kind (*ci*) (parent, older brother, husband) and loyal ministers are to be found. It is when fish forget the Dao of rivers and lakes that the virtuous act of moistening each other occurs. (Lynn 1999, p. 81)

This is a direct reference to *Zhuangzi* 6.2, which can be used to rebuke Confucian notions of reputation and role models, but equally, as in the hands of Wang Bi, strike a chord with core Confucian concerns. The relevant section of this passage reads:

> When the springs dry up, the fish have to cluster together on the shore, blowing on each other to keep damp and spitting on each other to stay wet. But that is no match for forgetting all about one another in the rivers and the lakes. Rather than praising Yao and condemning Jie, we'd be better off forgetting them both, letting their courses melt away in their transformation. (Ziporyn 2020, p. 56)

That fishes collect together and help one another survive is part of their natural disposition (*xing* 性). They do not learn to do this from others, nor is it even a matter of conscious effort. It is simply what they do so of their own accord, or an instance of "self-so." For the *Zhuangzi*, we can then use this to argue against relying on either positive or negative models and the focus on the reputation that is entailed therein. Rather than try to be like Yao, or protect oneself from being in any way similar to Jie, people are better off forgetting these models and returning to Dao. Wang Bi takes this a step further—or we

could say explicates what the *Zhaungzi* means more explicitly—by connecting this view to nearly all Confucian values. In this way, Chapter 18 is not about rejecting these virtues and values but rather states that, when they become the target of conscious intentions and actions, no longer represent genuineness (*zhen* 真). This is reminiscent of Confucius's own contempt for those hypocritical figures who falsely assume virtues, including Guan Zhong (3.22), the Ji family (3.1), and most famously (those who Mencius refers to as) "village worthies" (17.13).

This treatment of the chapter allows Wang Bi to be read as promoting "true" Confucian values while simultaneously not only not violating what is being said but actually drawing broadly on other sections of the *Laozi* and even the *Zhuangzi*. This is notably distinct from other early readings as well. Heshang Gong takes this passage to suggest that Confucian values can be useful for rehabilitating individuals and society in an effort to eventually return back to a "great Dao" state. For Heshang, then, there is still some separation between Confucianism and Daoism, at least insofar as their spheres of efficacy, respective values, and ideal states are concerned. For Wang Bi, they are simply making the same point in different ways—i.e., with different language, logic, and emphasis.

The main argument of this paper can be summarized by comparing Heshang Gong and Wang Bi's respective comments on chapter 18. While Heshang Gong makes space for separate Confucian values, Wang Bi does not take them as distinct. In Wang's work, both Confucianism and Daoism become transformed—but that is only if one first thinks of them as separate. According to his own understanding, "true Confucian philosophy" is completely commensurable with the *Laozi*. Insofar as both promote self-so and non-action, they are saying something similar, but as soon as virtues become the focus, they are not only distinct, but Confucianism is actually no longer a useful resource. In chapter 19, the *Laozi* itself, but especially in the comments of Wang, further supports this approach.

## 4. Simplicity and Decoration

Wang Bi's version of chapter 19 differs from some others, most significantly the Guodian text. Interestingly enough, with Wang, we have a sharper critique of Confucian values, but once again, his reading can be read as turning this to present the *Laozi* as completely in line with "true Confucian philosophy." However, as we will see below, there is one aspect that remains somewhat amiss. Wang's chapter 19 reads:

> Sever all ties with sagacity and give up wisdom, the people will be a hundred times better off. Sever all ties with humaneness and give up righteousness, the people will return to filial piety and parental care. Sever all ties with ingenuity and give up profit, there will be no more bandits and robbers. When these three things are used for adornment/decoration (*wen* 文), it will not suffice. Therefore, ensure that there is something to belong to: observe simplicity (*su* 素) and embrace plainness (*pu* 樸), be less concerned with yourself and minimize desires. (Chen 2020, p. 142; trans modified)

The Guodian has the first line as "sever all ties with wisdom and give up disputation." Chen Guying argues for the veracity of this line based on two points. Firstly, "sage" or "sagacity" is used as a metaphor for the highest state of personal cultivation. It is found thirty-two times in the *Laozi* and is nearly always highly praised. Already, then, one can be skeptical about this glaring exception in Wang's version. Secondly, "Sever all ties with sagacity and give up wisdom" appears twice in the *Zhuangzi* (11.2; 12.2). In both places, the *Zhuangzi* is discussing Confucian values in ways that are difficult to interpret as anything but harsh criticisms. Chen, therefore, believes that Wang's version is the result, once again, of later *Zhuangzi* extremists tampering with the *Laozi* (Chen 2020, pp. 142–43). In other words, when reading the first line, there is a huge discrepancy, and deciding on how to read this is pivotal for deciding on the "Confucian vs. Daoism (reading of the *Laozi*)" debate.[12] Discussion of the next line hinges on the same theoretical distinction.

Instead of "sever all ties with humaneness and give up righteousness," which directly rebuffs Confucian virtues, the Guodian version has "sever all ties with hypocrisy and

give up pretense." On the surface, this seems perfectly in line with Wang's thought—for instance, it is congruent with his understanding of chapter 18. Moreover, it smooths the edges of any sharp conflict between Confucianism and Daoism.[13] Additionally, many contemporary scholars unequivocally support the Guodian for textual and philosophical reasons, for example, Chen Guying,[14] Qiu Xigui 裘圭 (b. 1935),[15] Peng Hao 彭浩 (b. 1944),[16] Ding Yuanzhi 丁原植 (b. 1947),[17] and Yang Guorong 楊國榮 (b. 1957). Once again, however, Wang's reading almost requires this overt rejection of Confucian virtues in order to, paradoxically enough, prove that the *Laozi* supports them.

Wang Bi's full commentary to this chapter reads:

> Sagehood (*sheng*) and intelligence (*zhi*) designate the best (*shan* 善) of human talent (*cai*); benevolence (ren) and righteousness (*yi*) designate the best (*shan* 善) of human behavior (*xing*); and cleverness (qiao) and sharpness (*li*) designate the best (*shan* 善) of human resources (*yong*)! However, the text directly says that these should be repudiated. Because such "decoration" (*wen*) is utterly inadequate, one does not give people the chance to identify with these expressions and so never does anything that exemplifies what they mean. Thus the text says: Because these three pairs of terms serve as mere decoration, they are never adequate. When allowing people to identify with something, let them identify with your simplicity and minimal desires. (Lynn 1999, p. 82)

Always completely cognizant of his overall project, namely, the rejection of any possibility for constructing the means to develop institutionalized powers that eventually hollow out the true virtuosity of persons and societies, Wang calls instead for a total commitment to self-so and non-action; his comments here are best directed at Confucian values. In this way, the *Laozi* neither rejects them nor seeks to replace them with its own promotion of simplicity, plainness, and censure of selfishness and desires. There is no question that those latter ideas are promoted throughout the *Laozi*. However, when the above targets of severance and "giving up" are Confucian values, Wang can demonstrate how these values should be reinterpreted as self-so and non-action. They are thereby actually aligned with the ideas the *Laozi* constantly praises in a way that highlights an underlying congruence between the *Analects* and the *Laozi*.

Sagacity, wisdom, humaneness, and righteousness are not at all rejected. What is rejected—and Wang has already made this argument in chapter 18—is their use as mere decoration, which can also be broadly understood as criticizing them as "virtues." Thus, Wang not only furthers his project of rejecting falsity and any basis for pretense, which we find already in Confucius, but Wang also shows how the *Laozi* can be taken as supporting a certain reading of the *Analects* where self-so and non-action are recommended by both classics.[18] It does not reject these values themselves, but their mere appearance, and then reinterprets them. It rejects them in the way Confucius himself does. Of course, this was not successful, and so Wang Fu and Xu Gan had to repeat the argument. Wang Bi is now showing how this critique is shared by the *Laozi* as well. It, too, is against using sagacity, wisdom, humaneness, and righteousness as mere decoration. Instead, people should remain simple, plain, unselfish, and without desires. Cheng Xuanying 成玄英 (d. ca. 690) explains this as "returning to genuineness (*zhen* 真)"[19] (Lou 1980, p. 87). As was already shown in chapter 18, this means that the people will be *actually* humane and righteous. Real sages and true wisdom can emerge. Families will be harmonious, and that will have radiating effects on society. The *Laozi* does not refute this model; it opposes exactly what good Confucians should: the false appropriation of any aspect of this model, which can only harm persons and society. This eventually entails a complete censure of, as we will see in reference to chapter 38, *any* model and thus "virtues" themselves. Wang Bi himself spells out this point in his essay on the *Laozi*.

Right at the end of his essay, Wang basically provides a further explanation of his commentaries on chapters 18 and 19. He writes

Therefore a man of antiquity sighed and said: "Truly! What a difficult thing this is to understand! I knew that not being sagacious was not being sagacious, but I never knew that to be sagacious was not sagacious. I knew that not being humane was not being humane, but I never knew that being humane was not humane." However, thus it is that only after repudiating sagehood can the efficacy of sagehood be fully realized; only after discarding humaneness can the virtue of humaneness become really deep. To hate strength does not mean that one desires not to be strong but refers to how the conscious use of strength results in the negation of strength. To repudiate humaneness does not mean that one desires not to be humane but refers to how the conscious use of humaneness turns it into something false. (Lynn 1999, p. 40; translation modified)

Sagacity and humaneness do not simply get in the way of being "truly" sagacious or humane. This is an argument many "Confucians" bring to the text—such as David Hall and Roger Ames (cf. Ames and Hall 2003), Fung Yu-Lan (Feng Youlan) (Fung 1948) 馮友蘭 (d. 1990), and Fu Peirong 傅佩榮 (b. 1950) (Fu 2012), as will be mentioned below. Wang Bi goes much further. He seeks to replace not only trying to be these things but any degree of conscious and intentional action (especially those targeting specific values). Further, Wang does not aim to help one become the "true" version of sagacity and humaneness. Through casting them aside entirely, one can become efficacious and virtuous. In other words, by being self-so and practicing non-action, one will interact very well with others. The traces or footprints of these interactions are sometimes generalized and grouped into categories such as "sagacious" or "humaneness." Therefore, being self-so and practicing non-action does not ensure being the "true" versions of these virtues but being something else entirely. This is what others look back on and call instead "sagacious" or "humaneness," but that misses the point. It is all about being self-so and practicing non-action. As he writes, "Once the uncarved block (*pu*) fragments and genuineness is lost, all human affairs become permeated by villainy." (Lynn 1999, p. 38; translation modified). Comparing Wang Bi's comments on chapter 38 of the *Laozi* to other commentators further accentuates his unique reading.

## 5. Simplicity and Decoration

Chapter 38 is long, and Wang Bi's comments are rather extensive. Examining them closely rounds out our argument that Wang is not reading Confucian values into the *Laozi*, but rather exploring underlying similarities which do not significantly register as more "Confucian" than "Daoist." Comparing Wang Bi's reading to other interpretations makes this argument evident. Chapter 38 of the *Laozi* can be translated as:

Higher virtuosity (*de* ) is not virtuosity, and by this there is virtuosity; lower virtuosity does not let go of virtuosity, and by this there is a lack of virtuosity.

Higher virtuosity is non-action (*wuwei* 無為) and thereby does not depend on action; lower virtuosity acts for it and thereby has dependence on action.

Higher humaneness acts for it and is without dependence on action; higher righteousness acts for it and has dependence on action.

Higher ritual acts for it and there is no response, so sleeves are rolled up and things are cast aside.

Therefore, when the Dao is lost, there is virtuosity; when virtuosity is lost, there is humaneness; when humaneness is lost, there is righteousness; when righteousness is lost, there is ritual.

As for ritual, it is the thinness of loyalty and trustworthiness, and the head of disorder.

Those with foresight (appreciate) flower of Dao and (mark) the beginning of stupidity/duplicity.[20]

> Therefore the great person resides in the thick, and does not reside with the thin; resides in the actual (*shi* 實), and does not resides in flowery (*hua* 華). Thereby casting off that and taking up this. (Lou 1980, p. 93)

Even the decidedly Confucian scholars Roger Ames and David Hall have trouble defending a Confucian reading of this passage. There may be ways in which some congruence can be measured; "natural feelings" offer a bridge. However, overall, one cannot reject the obvious anti-Confucian elements. Ritual, ever important for Confucians, is expressly rejected. Ames and Hall write:

> This chapter joins the anti-Confucian polemic of chapters 18 and 19 in which there is real concern that the Confucian celebration of increasingly artificial moral precepts will overwhelm the unmediated expression of natural feelings. It is for this reason that the full arsenal of Confucian moral values comes under assault. (Ames and Hall 2003, p. 163)

Other readers of the *Laozi* who strongly favor similarities between it and the *Analects* often provide related explanations. There is something natural about Confucian values, and expressions of them are not a separate matter. As the *Analects* itself records, "adornment is like substance, and substance is like adornment (*wen you zhi ye, zhi you wen ye* 文猶質也，質猶文也)." (12.8). The *Laozi* is then read as a corrective on the growing focus on adornment; asking for a return to simplicity, genuineness, and innate dispositions means asking for a reemphasis on substance. Fung Yu-lan says this is how we can understand chapter 38—as a corrective against the imitation of mere decoration.

> As for the kind of humaneness and righteousness acquired through learning and training, these are always partly the result of imitation. In comparison to a naturally present genuine humaneness and righteousness, they are of a slightly lower order. When we read in the *Laozi* that "Higher *de* (virtue, efficacy) is not *de* (virtue, efficacy)" (chapter 38), this is what is meant. (Fung 1948, p. 128)

Imitation of Confucian values crowds out the potential for "unmediated expression of natural feelings." The assumption behind, and focus of, these more "Confucian" takes on the *Laozi* is that the person can be more in line with Confucianism when they do not become overly focused on decoration. People naturally tend to express virtues, which should be shaped, and the person cultivated, through learning from models and practicing virtues. However, when they fixate on appearances, true virtues suffer, and moral cultivation becomes learning to pretend. From this, trickery arises, and hypocrites emerge. On the surface, and in a truncated form, Wang Bi shares these views. In fact, his work was likely a huge influence—whether directly or indirectly—on these "Confucian" readings of *Laozi*. However, there is a critical difference. Wang Bi does not simply look for avenues of Confucian sympathy. In his long and detailed commentary on chapter 38, Wang mostly references "Daoist" ideas such as "emptiness," "non-action," "non-grasping," "not being selfish," and the like. Instead of simply overlaying Confucian ideas, he doubles down on Daoist commitments. Wang does not think that the virtues exposed by Confucians will "naturally" appear or become present in a "genuine" form. What will happen, however, is that behaviors and sentiments which are problematically labelled as such will manifest. Even appreciating them in this way, however, violates what Wang sees as the underlying argument about interacting well in Confucian and Daoist texts.

Wang Bi's explanation of chapter 38 is too long to quote in full, so we will instead summarize it. His commentary begins by noting the connection between virtuosity and Dao. Non-action is the full expression of them. Having an empty heart-mind (i.e., being without intentions) and being without selfishness and without person (*shen* 身) is how one can encourage harmony in those around them. Having virtuosity means "not regarding it as such, not holding to it, and not using it (*bu de qi de, wu zhi wu yong* 不其, 無執無用)." (Lou 1980, p. 93). "Thus, although (this person) has virtuosity (they) do not have the reputation for it" and it does not come to be part of their identity (Lou 1980, p. 93). Inferior virtue is the opposite. Additionally, reminiscent of chapter 2, Wang notes that as soon as "goodness

(*shan* 善)" is differentiated, its opposite arises. Invoking chapter 5, he says, "not acting for (certain names/reputation/differentiated values) means not having bias in action (*wu yi wei zhe, wu suo pian wei ye* 無以為者, 無所偏為也)." (Lou 1980, p. 93). Those who cannot do this concentrate on humaneness, righteousness, ritual, and etiquette. Therefore, acting for nothing is far superior to acting for something—no matter how much praise surrounds that something. Acting for nothing seems like acting for something, but it is not[21] "(*wu yi we er you wei zhi yan* 無以為而猶為之焉). (Lou 1980, p. 93).

Turning to the vocabulary that comprises his core thinking, Wang further explains that the "root (*ben* 本)" is non-action. If one focuses instead on action and acting for, then "although one might acquire a praiseworthy reputation, falsehood too will surely arise." (Lynn 1999, p. 121). This is echoed politically as well. The more one tries to straighten others and make them sincere, the more problems will arise. Confucian values become superficial not so much because the decorative aspects are over-emphasized, but even in trying to promote "natural feelings" and "genuineness" do "cultural institutions and ceremonial etiquette become superficial ornamentation." (Lynn 1999, p. 121). Holding to Dao, through non-action, is the only viable alternative. "Even if humaneness and righteousness emerge from within, acting from (for) them is still like pretense and falsity (*fu ren yi fa yu nei, wei zhi you wei* 夫仁義發於, 為之猶偽)." (Lou 1980, p. 94). Here Wang Bi differs wildly from Fung, Fu, and Ames and Hall. His "Confucian" *Laozi* must not be so quickly labeled as such. Virtues, even those which are entirely "genuine" or "natural and unmediated expressions of true feelings," are equated with falsity.

Compared with the discussion of this chapter in the *Hanfeizi* 韓非子 (Book of Master Hanfei), Wang's unique contribution can be even further highlighted. The *Jie Lao* 解老 or "Explaining the *Laozi*" section of the *Hanfeizi* begins with an extensive reading of Chapter 38—nearly 100 characters longer than Wang's over 1200 character commentary. Here the *Hanfeizi* makes constant references to "actualities (*shi*)," "emotions (*mao* 貌)," and "principles/patterns (*li* 理)." The overall argument is that people do not fully appreciate Confucian virtues and the role of ritual. The virtues[22] are formalized expressions of "actual emotions." The *Hanfeizi* comments: "Humaneness is to happily love others from your inner heart. It is to delight in others' good fortune and to detest others' misfortune." (Queen 2013, p. 229). Rituals come from the person (*shen* 身) and are supposed to be an accurate expression of their emotions. In and of themselves, humaneness and ritual are not problematic. Rather, to the contrary, the *Hanfeizi* promotes them in a manner that has led some to call it a synchronism of Confucianism and Daoism.[23] The discussion of chapter 38 in the *Hanfeizi* ends:

> The expression "rejects the one and appropriates the other" refers to disregarding the outer appearance of ritual and random guessing and abiding by the causes of things in the ordering principles and the inner substance of the emotions (*qing shi*). Thus it is said: "(He) rejects the one and appropriates in the other." (Queen 2013, p. 231)

Wang Bi does not interpret the "actualities (*shi*)," "emotions (*mao* 貌)," "principles/ patterns (*li* 理)," or the "person (*shen* 身)" in the same way at all. For him, any discussion of "virtue" already violates whatever may be valid in the abovementioned concepts. This becomes fleshed out even more clearly in the (roughly) last third of his commentary on chapter 38.

Discussing how those with foresight only appreciate the "flower (*hua* 華)"[24] of Dao and thereby mark the beginning of stupidity/duplicity (*yu* 愚)[25], Wang says that even if they get at the natural tendencies (*qing* 情) of things and situations, everything will eventually be worse off because of it. Reputation and other superficial gains will be had at the cost of sincerity (*du* 篤) and honesty/the actual (*shi*). It is best to let go of one's self and go by things—this is how non-action leads to peace "(*she ji ren wu, ze wuwei er tai* 舍己任物, 則無 為而泰)."[26] (Lou 1980, p. 94). Holding to the "mother"—or uncarved block/simplicity (*pu*) as Lou Yujie reads it (Lou 1980, p. 104)—one can effectuate positive change and manage

affairs well. Ultimately, Lou explains, this means acting in a self-so and non-action manner. The connection is made by Wang himself already in his essay on the *Laozi* (Lou 1980, p. 104).

The "mother" or uncarved block/simplicity (*pu*) and self-so and non-action are the focus for Wang. From them, actions and sentiments, which we may refer to as humane or righteous, and even ritual and etiquette, are manifest. However, these labels are themselves always problematic and should never be the target. They are born from the mother (as self-so and non-action), but when we focus on them, we completely lose sight of how they came to be in the first place. Therefore, while we may use language to discuss them and make distinctions, we should never rely on these things, lest we undermine the importance of the mother in favor of the children. Wang writes: "It is because one functions not by using forms and rules and not by using names that it becomes possible for humaneness and righteousness, propriety and etiquette to manifest and display themselves." (Lynn 1999, p. 123; translation modified).

Wang's argument is not simply that the appearance of virtues has taken the place of their true, natural, or genuine versions. This is characteristic of a Confucian reading: displaying a clear commitment to values associated with the *Analects*, *Mencius*, and *Xunzi*. These readings take the *Laozi* to be complaining merely about the misappropriation of these values. Central concepts such as self-so and non-action are the more "unhewn," plain, and simple or "stripped down" versions of humaneness, righteousness, propriety, and etiquette. Wang's philosophical interpretation is significantly different. Major notions in the *Laozi*, including self-so, non-action, not being selfish, and being without desire, are themselves the best version of human sentiment and behavior. Confucian labeling and distinctions are problematic because they redirect attention away from these ideals. There is still overlap with Confucius, and he himself demonstrated some of the ideas found in the *Laozi* even better than the *Laozi* discusses them, but the ideals themselves remain Daoist.[27]

## 6. Conclusions

Despite his own reverence of Confucius as a higher sage than Laozi, Wang Bi would likely gawk at any classification of his reading of the *Laozi* as "Confucian." Insofar as the *Laozi* and *Analects* promote self-so and non-action, while simultaneously rejecting selfishness and desires, they are saying something similar. Their underlying projects are meaningfully commensurable. However, as soon as we introduce "Confucianism" as a school of thought, which almost invariably means transforming it into a "teaching of names" and with this the all but necessary requests of imitation and a general neglect of the "root" or "mother," Wang Bi must be taken as vehemently arguing against this school. What he does agree with is Confucius as an exemplar of praising non-being—and in terms of practice, this means self-so and non-action. If, however, Confucius' teachings, which are completely person- and situation-specific, become theorized as more abstract virtues, then the underlying message (what Wang calls "that by which" or *suo yi* 所以) is ignored. Taking Wang Bi as a mere commentator overlooks his deeper philosophical understanding.

The nuance and complexity of his unique philosophical appreciation of the *Laozi* are often oversimplified and understood as merely harmonizing, synthesizing, or unifying Confucianism and Daoism. However, Wang Bi is better appreciated as a philosopher in his own right, one who draws heavily from a variety of texts and sees similarities in the general trajectory of Confucian and Daoist philosophies. Like many Confucians (and nearly all systems of thought, ethical theories and the like), Wang Bi is opposed to pretenders and hypocrites. Those who merely take on the names, are after reputation, or only get at the "flower of Dao", are criticized. However, this does not make Wang Bi Confucian. To be sure, he rejects the mere appearance of virtuosity because it gets in the way of "true" virtuosity, but this virtuosity is not Confucian virtue, and must be understood as ultimately self-so and non-action. The ways we talk about self-so and non-action often refers to convenient labels such as humaneness and righteousness; however, Wang Bi is not interested in them as such. This is clear in Wang's summary of the *Laozi*: "The *Laozi* can be almost completely covered with a single phrase: Ah! It does nothing more than encourage the growth at the

branch tips through enhancing the roots." ([Lynn 1999](), p. 37; translation modified). As Lou argues, the roots are the mother, or Dao, or self-so and non-action ([Lou 1980](), p. 104).

The implications of how we understand Wang Bi's work are far-reaching. Not only has his version of the *Laozi* been dominant for most of Chinese history, but his commentary, too, has been inestimably influential. Countless interpretations and hundreds of translations have relied heavily on Wang Bi. This paper contributes to the "Global Laozegetics" project in two at least major ways: firstly, by reinterpreting Wang Bi's own reading of the *Laozi*, and second, through addressing how other translations and interpretations are affected.

**Funding:** This work was funded by the Fundamental Research Funds for the Central Universities [No.: 2018ECNU-QKT010].

**Institutional Review Board Statement:** Not applicable.

**Informed Consent Statement:** Not applicable.

**Data Availability Statement:** This study did not report any data.

**Conflicts of Interest:** The author declares no conflict of interest.

## Notes

1.  Given Wang Bi's approach, which is the dominant theme of this paper, I will use this type of translation for all "masters texts."

2.  Wang Bi has long been classified as a *Xuanxue* thinker—*Xuanxue* was translated by Fung Yu-lan as "Neo-Daoism" into English, which resulted in Wang's being labeled as a "Neo-Daoist." Reading Wang Bi as heavily invested in "Confucian" values is the result of exploring exactly what *Xuanxue* or "Neo-Daoism" means in a more nuanced sense. In other words, scholars who provided more nuanced accounts of Wang Bi describe his *Xuanxue* as "Confucian."

3.  Here Ziporyn is actually speaking about Wang Bi's great philosophical successor Guo Xiang 郭象 (d. 312), but he attributes this same project to Wang Bi. The quote ends "even more thoroughgoingly than any of his predecessors had."

4.  Nelson provides an excellent account of the various positions on Wang Bi and Confucianism ([Nelson 2020](), pp. 288–90). For a fuller discussion see [Yang]() ([2010]()); [Ziporyn]() ([2020]()); and [Wang]() ([1987]()).

5.  Misha Tadd provides a philosophical investigation of the Heshang Gong commentary see [Tadd]() ([2020]()).

6.  For more on these terms see [Chen]() ([2020]()) and [Moeller]() ([2006]()). Further detail on Wang Bi's own interpretation of these terms are developed in the body of this paper, see also [Chan]() ([2018]()) and [D'Ambrosio]() ([2019]()). This entire essay is a specific elaboration on much of D'Ambrosio's argument in [D'Ambrosio]() ([2019]()).

7.  For further discussion of these meetings see [Chan]() ([2010]()), they are also widely discussed in all the Chinese language references to Wang Bi or Xuanxue in this paper.

8.  Much of Wang's commentary to the *Analects* is lost. His commentary to the *Book of Changes* was the official and standard reference until the Song dynasty, when Zhu Xi's 朱熹 (d. 1200) work became dominant.

9.  The "anti-Confucian" element is plain enough, however, for some examples of this reading see [Moeller]() ([2004](), [2006]()).

10. There are slight discrepancies in some of the characters in The *Guodian jian* 郭店簡 [*Guodian Bamboo Slips*] version, the *Boshu* 帛書 [*Mawangdui Silk Manuscript*] versions, and the Fu Yi 傅奕 (d. 639) version (cf. [Chen 2020]()). Here we are mainly concerned with Wang Bi's version, and his reading of the text. In any case, the differences in these versions is minimal and does not greatly affect a philosophical reading. See ([Chan 2018]()). [https://plato.stanford.edu/entries/laozi/](https://plato.stanford.edu/entries/laozi/) (accessed on 13 February 2022). An exception will be address below.

11. Moeller is implicitly referencing the paradoxical logic or thinking of the *Laozi*, expressed most poignantly in chapter 2 of the *Laozi*.

12. These chapters (5, 18, 19, 38) are among the most contested in the entire *Laozi*. Those who find affinity between the *Laozi* and the *Analects* (e.g., Ames and Hall, Chen Guying) promote a certain reading, while those who emphasize a difference between them read these chapters as criticizing values associated with Confucianism (e.g., Fu Peirong, Moeller). For a detailed discussion of this debate see [D'Ambrosio]() ([2022]()).

13. As mentioned in the previous footnote, there are various ways to interpret this chapter. Moeller, as opposed to Wang Bi, emphasizes conflict between "Confucianism" and "Daoism": "the chapter continues and even amplifies the anti-Confucian polemics of the two preceding chapters. The first three sentences [ . . . ] denounce the Confucian virtues and ask for their elimination. The Confucian virtues are seen as obstacles to the creation of a good society. Rather than bettering the world, they contribute to the evils they are supposed to remedy." ([Moeller 2007](), p. 71).

14. See [Chen]() ([2020]()).

15. '*Guodian Chumu zhujian*' *zhushi* 郭店楚墓竹簡:註釋 [*Annotated Edition of the Bamboo Slips in the Chu Tomb at Guodian*] (cf. [Chen 2020]()).

16   *Guodian Chujian Laozi jiaodu* 郭店楚簡老子校讀 [*Collation and Reading of the Guodian Chu Bamboo Slips of the Laozi*] (cf. Chen 2020).

17   *Guodian zhujian Laozi shixi yu yanjiu* 郭店竹簡老子釋析與研究 [*An Analysis and Investigation into the Guodian Bamboo Slips of the Laozi*] (cf. Chen 2020).

18   Wang's philosophy does not assume a dating for the *Analects* or the *Laozi*. Both are referenced as resources for reflecting on philosophical topics. In short, he is not engaged with what we today could call "academic philosophy."

19   Translations not otherwise indicated are my own.

20   The character being translated is "*yu* 愚" most read this as "stupidity." There is also reason to read it as duplicity (cf. Lynn 1999, p. 119).

21   This is not a translation of the quote that follows, but a rough reading of its general gist.

22   In this context it is more accurate to speak of them as "virtues." For Wang Bi "values" is a more appropriate designation.

23   Sarah Queen, for example, writes: "Perhaps one of the most striking characteristics of "Jie Lao" that set it apart from "Yu Lao" is its syncretic quality. "Jie Lao" seeks to harmonize practices and ideas that later became associated with the "Daoist" and "Confucian" traditions, a quality not present in "Yu Lao." Moreover, the syncretism of "Jie Lao" appears to be devoid of influence from what later became identified as "Legalism." The commentary does not discuss typical "Legalist" notions of governance such as rewards and punishments, names and actualities, the importance of political purchase or impartial laws." (Queen 2013, p. 212). This reading of the "Jie Lao" chapter is well supported. The text waxing on about the positive role of the Confucian virtues and practices explicitly rejected in chapter 38. Wang Bi does not, making the classification of his commentary as "pro-Confucian" or even syncrenist a gross over simplification. His project is far more complex than what we find in "Jie Lao" or in Fung, Fu, or Ames and Hall.

24   The "flower" refers to decorative aspects, or those ornamental qualities that are not directly related to the "substance" of Dao.

25   See note 18 above.

26   This sentiment is echoed and rephrased throughout Wang's commentary. In the conclusion we will revisit his work on chapter 5 to see one particular reiteration.

27   Again, this does not assume Confucius did or did not read the *Laozi*, or any dating of either.

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
