# Peer review of "Wang Bi’s “Confucian” Laozi: Commensurable Ethical Understandings in “Daoist” and “Confucian” Thinking"

_religions, doi:10.3390/rel13050417_

Round 1

Reviewer 1 Report

The paper makes a strong argument against reading Wang Bi’s interpretation of the Daodejing as Confucian. In general, the paper is very good, but there are few things that can be rethought in its execution.

The author suggests that the normal identification of Wang Bi’s thought is as Confucian, and that is what a good part of the paper’s argument concerns. “In terms of ethics, virtues, and values Wang Bi’s 王弼 (d. 249) work on the Laozi is normally read as mainly “Confucian.” Still, it’s mostly scholars with an affinity to Confucian philosophy that consider him as a Confucian, but he is also often considered either as a neo-Daoist thinker or as a Xuanxue thinker, and while the author alludes to this, it can be clarified, otherwise the reader gets the wrong impression of the different ways in which his philosophy has been approached. On p. 8, the author mentions the “Confucian vs. Daoism (reading of the Laozi)” debate, but the Daoism reading does not get much representation, so this could be better represented.

The beginning of the paper, on p. 3, states, “The overall argument is that while Wang’s work has been evaluated as a “Confucianization” or “Confucian-based synthesis,” there is actually no clear evidence that this is a conscious project pursued by Wang Bi himself. It is just as likely, and perhaps even better evidenced, that Wang’s reading represents an accurate account of the Laozi itself.” I have no idea what it means that “Wang’s reading represents an accurate account of the Laozi itself,” and this sentence is not beneficial for the reader trying to follow the paper’s main argument. Leaving aside the question of whether Wang Bi is somehow perfecting the thought of the Laozi, it nevertheless takes the author a while before clearly and definitively setting forth the main thesis, that Wang Bi’s thought ought not to be considered Confucian. For much of the paper, the author has not decisively stated this, as seen in the line quoted from p. 3 above, but the conclusions in the final section do so. Before then, the reader is often wondering where the author is going, but if the first section more decisively lays out the final destination, the reader can more comfortably follow the argument. Thus, the author can consider setting forth a very clear statement in the beginning of the paper along the lines that Wang Bi is not improving or perfecting the Daodejing, that he is not offering a Confucian interpretation, and that he is not offering a Daoist interpretation; rather, he is offering his own interpretation as neither Daoist nor Confucian. Is Wang Bi articulating a Xuanxue interpretation, or is it just a Wang Bi interpretation that should not be classified in relation to any -ism?

Sometimes the author insinuates that the Laozi was written after the Analects; on p. 8, for example, s/he writes “Wang also shows how the Laozi supports a certain reading of the Analects where self-so and non-action are recommended by both classics.” Sometimes the author insinuates the reverse; on p. 13, for example, the author writes, “Confucius himself demonstrated some of the ideas in the Laozi even better than the Laozi discusses them, but the ideals themselves remain Daoist.” The author can think about clarifying his/her position on the actual (whether ‘historical’ or ‘textual’) relation between Laozi and Confucius outside of Wang Bi, because doing so would allow the reader to follow the argument concerning Wang Bi more clearly.

There is nothing wrong with the main arguments, in fact the author is producing an excellent contribution to the field of Chinese philosophy, but if the issues I have raised can be ironed out, then it will allow the philosophical content to be more immediately presented in a clear and unambiguous manner. Which brings me to the final comments, that the author has a lot invested especially in these three terms: wen/ decoration, wuwei/ non-action, and ziran, self-so. If the author can introduce these concepts and how s/he is using them, it would also be helpful for the reader.

Author Response

The second reviewer did not ask for changes so here I respond to Reviewer #1:

Paragraph #1:

The reviewer makes and excellent point about understandings of Wang Bi as a Xuanxue or Neo-Daoist thinker. I have added three sentences to clarify this in the introduction. I’ve also added some remarks in the main body of the text.

In response to the point about “Confucian vs. Daoist” readings of the Laozi I have further clarified this with several sentences and by citing two additional sources.

Paragraph #2:

I want to sincerely thank the reviewer for pointing out this issue. The paper changed significantly in writing it, which became well state in the conclusion. However, the introduction and first couple of pages do not accurately reflect what came to be the main focus of the paper. Thank you again. I have changed the introduction and first pages to make my argument more clear. How to classify him is also now addressed.

Paragraph #3

Wang Bi’s philosophy reads the Analects and Laozi as resources. The dating of these texts in relation to one another is not important for him. I have added notes discussing this. As for my own position I do not think it is relevant for the paper, though I have added notes to show Wang’s position.

Paragraph #4

The reviewer is correct to point out that this essay relies heavily on “decoration,” “non-action,” and “self-so” without providing an introduction these concepts. They are outlined in the introduction, on page three, but not specifically spelled-out. I have added to the introduction, as well as a note and another source to address this issue.

Reviewer 2 Report

This is an excellent essay! Well-researched and argued. It includes a good amount of primary materials and secondary scholarships. It will help the field to get into a deeper understanding of Wang Bi's interpretations of Laozi and avoid a defect of the oversimplification of Laozi.

In the lines of 354-355, there is a list of Chinese scholars' with their birthdays. Is this necessary? It seems there is no sound reason for that kind of information.

Author Response

Thank you!

I listed their dates just to give context. If the journal would like me to take them out I happily will!